# Torticollis in Connection with Spine Phenotype

**DOI:** 10.3390/diagnostics12071672

**Published:** 2022-07-09

**Authors:** Ali Al Kaissi, Nabil Nassib, Sami Bouchoucha, Mohammad Shboul, Franz Grill, Susanne Gerit Kircher, Polina Ochirova, Sergey Ryabykh

**Affiliations:** 1National Medical Research Center for Traumatology and Ortopedics n.a. G.A. Ilizarov, 640014 Kurgan, Russia; poolen@yandex.ru (P.O.); rso_@email.ru (S.R.); 2Department of Paediatric Orthopedic Surgery, Children Hospital, Tunis 1029, Tunisia; nabil.nassib@rns.tn (N.N.); sami.bouchoucha@yahoo.com (S.B.); 3Department of Medical Laboratory Sciences, Jordan University of Science and Technology, Irbid 22110, Jordan; mohammad.shboul@reversade.com; 4Pediatric Department, Orthopedic Hospital of Speising, 1130 Vienna, Austria; grill.franzleo@gmail.com; 5Department of Medical Patho-Chemistry and Genetics, Medical University of Vienna, 1090 Vienna, Austria; susanne.kircher@meduniwien.ac.at

**Keywords:** torticollis, spine phenotype, radiology, tomography, syndromic association, genotype

## Abstract

Purpose: Torticollis is not of uncommon occurrence in orthopaedic departments. Various theories and studies concerning the pathogenesis of the deformity have been suggested. We aimed to highlight and discuss the underlying cervical and spine malformation complex in correlation with torticollis via radiographic and tomographic analysis and its connection with a specific syndromic entity. Methods: Torticollis has been recognised in six patients (2 boys and 4 girls with an age range of 14–18 years), in addition to a couple of parents manifested persistent backpain. A variable spine malformation complex was the main reason behind torticollis. In addition, some patients manifested plagiocephaly, facial asymmetry and scoliosis/kyphoscoliosis. In some patients, conventional radiographs were of limited value because of the overlapping anatomical structures. Three-dimensional reconstruction CT scanning was the modality of choice, which enlightens the path for the phenotypic characterisation. Results: A 16-year-old-boy presented with torticollis in correlation with pathologic aberration of the spine cartilaginous stage was analysed via 3DCT scan. Comprehensive clinical and radiological phenotypes were in favour of spondylomegepiphyseal dysplasia. The genotype showed a mutation of the NKX3-2 (BAPX1) gene compatible with the diagnosis of spondylo-meg-epiphyseal-metaphyseal dysplasia. His younger male sibling and parents were heterozygous carriers. In two patients with pseudoachondroplasia syndrome, in which odontoid hypoplasia associated with cervical spine synchondrosis causing life-threatening torticollis, Cartilage oligomeric matrix protein (COMP) gene mutation was identified. MURCS syndrome has been diagnosed in two unrelated girls. Torticollis associated with cervical kyphosis was the major presentation since early childhood. Interestingly, one girl showed omovertebral bones of the lower cervical and upper thoracic spine. Her karyotype manifested a balanced translocation of 46 XX, t (14q; 15q). Conclusion: To detect the underlying etiological diagnosis of torticollis, a skeletal survey was the primary diagnostic tool. Conventional radiographs of the craniocervical junction and spine resulted in confusing readings because of the overlapping anatomical structures. Cranio-cervical malformation complex could have serious neurological deficits, especially for children with indefinite diagnosis of torticollis. The widely used term of congenital muscular torticollis resulted in morbid or mortal consequences. Moreover, some patients received vigorous physical therapy on the bases of muscular torticollis. Sadly speaking, this resulted in grave complications. Understanding the imaging phenotype and the genotype in such patients is the baseline tool for precise and proper management. The value of this paper is to sensitise physicians and orthopaedic surgeons to the necessity of comprehensive clinical and radiological phenotypic characterisations in patients with long term skeletal pathology.

## 1. Introduction

Tubby [1] was the first to define torticollis as a deformity (congenital or acquired in origin characterised by lateral inclination of the head to the shoulder, with torsion of the neck and deviation of the face. Congenital muscular torticollis (CMT) is the most common type of torticollis, a condition that begins in infancy as a rotation and flexion deformity of the neck caused by sternocleidomastoid muscle (SCM) shortening.

Spondylomegaepiphyseal-metaphyseal dysplasia (SMMD; OMIM 613330) is a rare skeletal dysplasia that was first described by Silverman and Reiley [2] in eight patients with a condition affecting the spine, epiphyses and metaphyses of the long bones. The trunk and neck were short (the neck was also stiff) and there was defective ossification of the vertebral bodies with the anterior halves being separated from the well-developed posterior elements. In addition, the thorax was short and broad, and the limbs were relatively long with contractures. The causative gene was first reported by Hellemans et al. [3]. They identified two homozygous loss of function variants in the NKX3-2 gene (OMIM 602183) in four affected children from two consanguineous Turkish and Moroccan families.

Pseudoachondroplasia is characterised by short stature, which may have passed unnoticed at birth, with the vast majority of children manifesting achondroplasia at the age 2–3 years. At birth, there were no craniofacial dysmorphic features to be detected. The maximal height of the adults is usually between 90 and 145 cm [4,5].

MURCS association (OMIM 601076) also referred to Müllerian aplasia (MA) as a rare constellation of anomalies that include uterine aplasia/hypoplasia, renal agenesis/ectopy, abnormal cervical or upper thoracic vertebrae, abnormal ribs, Sprengel shoulder, upper limb abnormalities and deafness [6]. Thus, patients who have features of spine malsegmentation with Rokitansky/urogenital dysplasia syndromes (OMIM 277000) are included within the diagnosis of MURCS syndrome. The genetic cause of MA has been well-studied. Mutations in the LIM homeobox 1 (*LHX1*), and wingless-type MMTV integration site family, member 4 (*WNT4*) have been identified to be causative of MA [7,8].

The malformation complex of the craniocervical junction are of great concern to physicians and spine surgeons [9]. Craniocervical instability/deformity could initiate serious fatal/morbid outcome for children, especially if vigorous physical therapy is to be organised [10]. The clinical appearance of children with cranio-cervical abnormalities is usually seen in a long list of syndromic entities such as Goldenhar syndrome [11]. Frankly speaking, conventional radiographs of the craniocervical junction are often difficult to read in young children and infants and may be confusing. In practice, spine malformation is associated with a wide spectrum of skeletal dysplasia and syndromic associations. We meticulously scrutinise our patients via clinical and radiographic phenotypic characterisation. 3D reconstruction CT scan is a mandatory tool to fully understand the reason behind fixed torticollis.

New technologies such as next-generation sequencing and microarray technologies and animal modelling can facilitate finding the correct diagnosis but can unfortunately not replace the clinical and radiological phenotypic delineation and elucidation.

## 2. Materials and Methods

The study protocol was approved by the Ethics Committee of the Ilizarov Scientific Research Institute, No. 4(50)/13.12.2016, Kurgan, Russia. Informed consents were obtained from the patients’ guardians. This study was conducted based on a clinical and radiographic evaluation of a group of children who presented with torticollis. The radiological examination consisted mainly of anteroposterior (AP) radiographs, lateral cranio-cervical (the head was held as straight as possible) and AP spine radiographs. Previous reports, investigations and surgical interventions have been reviewed by the senior author and were discussed thoroughly with the colleagues. Patients with torticollis were documented in our Institute on bases of detailed clinical and radiological phenotypic characterisations.

Torticollis was the main clinical presentation. The clinical phenotypic characterisation showed fixed torticollis with a deviation of the head. Patients were unable to perform any head movements by themselves and, in an attempt to turn the heads of the patients, we were incapable to move past the midline. To ascertain the aetiology of torticollis, clinical and radiological phenotypic characterisations were used as the baseline tools.

Patients were subdivided in accordance with the definite diagnosis.

## 3. Torticollis in Connection with Pathologic Aberration of the Spine Cartilaginous Stage in A Family with Spondylomegepiphyseal Dysplasia

A 16-year-old boy, with Marfanoid habitus, long and thin limbs and dysmorphic features. Torticollis associated with kyphoscoliosis were the major clinical presentations. He was born to a 21-year-old mother and a 27-year-old father after a pregnancy of 40 weeks gestation and normal delivery. The parents are related. Birth weight, length and OFC were all around the 50th centile. His motor skills were delayed. Torticollis was the major deformity for which parents sought advice. Lengthening of the sternocleidomastoid muscles was performed in his first year of life in an attempt to correct the deviation. Unfortunately, no good results have been encountered and his torticollis persisted. Simultaneously, he developed abnormal gait when he started to walk. His mental development was normal. On physical examination, his height was 155 cm (3rd centile), his weight 51 kg (10th centile), and his head circumference 54 cm (50th centile). There was a noticeable restriction of neck movements, and the head was tilted towards the left side. Pertinent physical findings were generalised as ligamentous hyperlaxity, relative arachnodactyly, valgus deformity of the big toe, pectus excavatum and cervico-thoracic kyphoscoliosis. Facial abnormalities included hypertelorism, maxillary hypoplasia, prominent jaw, pointed chin and small mouth. Vision and hearing were normal. Cardiac, abdominal and renal ultrasound was normal. Biochemical investigations of the proband including routine blood and urine examinations, urine amino acids and chromatography, mucopolysaccharide screening test and karyotype (Fish), were all normal.

Reformatted CT scan of the craniocervcial and the cervical vertebrae of the proband showed odontoid hypoplasia secondary to neurocentral synchodrosis and defective formation of the cervical vertebrae in connection with synchondrosis. The latter is the development of union between two bones by the formation of either hyaline cartilage of fibrocartilage. A synchondrosis is usually temporary and exists during the growing phase until the intervening cartilage becomes progressively thinner during skeletal maturation and is ultimately obliterated and converted into bone before adult life. In other words, a synchondrosis is a cartilaginous joint. It allows only slight movement between bones compared to the synovial joint, which has a much greater range of movement. The process of ossification within the centrum of the vertebral body is similar to that of tertiary ossification. Longitudinal growth mostly occurs at the chondro-epiphyseal portions of the end-plates. In this patient, it was obvious that synchondrosis was a permanent rather than a temporary process. Reformatted Coronal CT scan of the cervical spine showed butterfly vertebrae (defective formation), note the detached cephalic part of the odontoid process (arrowhead) in connection with extensive cervical spine synchondrosis causing the mal-development of butterfly vertebrae (Figure 1a). Reconstruction CT scan of the cervical spine showed atlanto-axial dislocation (arrows). Both parents experienced low backpain since their late adolescence (spondylolisthesis was the diagnosis). AP radiograph of the lumbar vertebrae at the age of 32-year-old mother showed osteoarthritis of the facet joints of L5 (arrow) associated with diminution of the heights of L4/5. Note the increased level of calcification that signifies facet arthritis. Surprisingly, the AP pelvis radiograph of the mother at age of 32 years showed incidental diagnosis of ossification of the abdominal aorta. The mother is asymptomatic, but such finding is of utmost importance. Ossification of the abdominal aorta signifies a state of subclinical atherosclerosis with subsequent vascular hazards-red arrowhead (Figure 2a). Sagittal spine MRI of the 38-year-old father showed dysplastic spondylolisthesis of L4/5 (arrow) that might leads to spinal stenosis (Figure 2b). Phenotype/genotype of the family with spondylo-meg-epiphyseal-dysplasia appears in Table 1.

## 4. Results

A sequence analysis of the NKX3-2 (BAPX1) gene revealed a homozygous 19 base pairs duplication (c.-2_17dupAGATGGCTGTGCGCGGCGC) in the proband. The parents were heterozygous carriers. The duplicated region includes the initiation codon, which could lead to the production of a truncated protein. The identified variant has not been previously reported in the databases nor in the literature.

## 5. Torticollis Associated with Odontoid Abnormalities and Atlanto-Axial Instability in Patients with Pseudoachondroplasia

Our patients manifested short stature associated with distinctive bone changes along the acetabulo-femoral, metacarpal and the metaphysis of the weight-bearing joints. At the craniocervical junction, persistence of axial synchondrosis was apparent. Extensive and unusual pronounced acetabulo-femoral changes were seen. The hands were small, stunted and showed proximally pointing metacarpals. Significant metaphyseal striations around the knees were notable. The overall phenotype and genotype were consistent for the diagnosis pseudoachondroplasia.

AP cervical and upper thoracic spine of a 16-year-old girl with pseudoachondroplasia showed torticollis associated with unusual flattening of the cervical spine. The MRI has illustrated the remnant of the persisting dentocentral synchondrosis. The localisation and level of the remnant of the dentocentral synchondrosis, started from the detached odontoid process and extended downwards to involve the entire cervical spine (as part of a generalised cervical spine synchondrosis (Figure 3a,b). Not only is this anomaly an important anatomical detail, but it is also very relevant from a clinical perspective because of the odontoid and C2 fractures.

A 14-year-old girl manifested the phenotype and genotype of pseudoachondroplasia, presented with torticollis. Radiological and tomographic studies confirmed the diagnosis of os odontoidium.

Three-dimensional CT scan showed the round type of os odontoideum which predisposes to myelopathy. 3D reconstruction coronal CT scan of a 14-year-old girl with pseudoachondroplasia showed rounded shape os odontoideum (arrow) and atlanto-axial instability (Figure 4a). Three-dimensional sagittal reconstruction CT showed exaggerated cervical spine lordosis (Figure 4b).

## 6. Results

Sequence trace analysis revealed that the 16-years-old-girl was heterozygous for the nucleotide change c.950A > G in exon 9 of the *COMP* gene. This mutation is predicted to result in the amino acid substitution p.Asp317Gly in the 1st type III repeat of *COMP* (T3_1_). The 14-year-old girl showed mutations in *COMP* gene on chromosome 19 p13.1-p12.

## 7. Torticollis in Patients with MURCS Association

An 18-year-old girl with under-developed secondary sex characteristics, but primary amenorrhoea and cervical vertebral defects can raise our doubts for the diagnosis of MURCS. She showed enormous alterations in the normal sequence of development of Mullerian ducts which resulted in the development of a wide spectrum of reproductive tract abnormalities. Pelvic and genital ultrasound examination revealed vaginal atresia and a rudimentary uterus, which was confirmed by the colposcopy examination showing symmetrical muscular buds instead of a normal uterus, and renal ultrasound showed normal kidneys. The clinical examination revealed a normal growth spurt, but torticollis and cervical kyphosis were the most bothersome complaint. She received abundant sessions of physiotherapy in her first few years of life. The clinical examination showed a short neck with inclination of the head to the same side (fixed torticollis) strongly correlated with cervical kyphosis. She totally lost her spine biomechanics and stiffness was the outcome. Three-dimensional sagittal CT scan showed cervical kyphosis with massive fusion of the cervical vertebrae (extensive fusion of C4–7 associated with fused spinous processes), a deformity that is extremely surprising (Figure 5a). Axial 3D reformatted CT scan of the C7 showed a bony bridge between the omoplate and spine which resulted in the development of an omovertebral bone (Figure 5b).

A-16-year-old girl similarly presented with torticollis since her early life followed by the development of progressive cervico-thoracic kyphosis. Pelvic-renal ultrasound showed a constellation of ovarian aplasia, uterine, vaginal hypoplasia associated with left renal hypoplasia. Thus, the patients have a combination of Mullerian duct aplasia, renal hypolasia, and cervicothoracic somite dysplasia. Three-dimensional reconstruction CT scan in a 16-year-old girl showed torticollis, progressive cervico-thoracic kyphosis, and bilateral omovertebral bones originating from the 7th cervical spine. An apparent omovertebral bone connecting the 7th cervical spine with the superior border of the left scapula (white arrow) gives rise to the development of Springle’s deformity (Figure 5c).

## 8. Results

The 18-year-old girl underwent a chromosomal analysis showing a balanced translocation of 46 XX, t (14q; 15q). The parents of the 16-year-old girl rejected any additional laboratory or genetic testing.

## 9. Discussion

The entire back bone segments are ossified from two centres, but the atlanto-axial vertebrae are ossified from three ossification centres. There is a steady and progressive union from these three ossification centres. The union of the primary ossification centres of the anterior centrum and posterior arches occurs gradually. A major part of the anterior portions of the neural arch contribute to vertebral body formation. The anatomy of the atlas (C1) is distinctive; it shows that the anterior primary ossification centre is present in fewer than 20% of all new-born infants. The anterior ossification is termed as the intercetraxium-1. The posterior synchondrosis and the anterior neurocentral synchondroses are fused by the time a child is 4 or 5 years old, and the diameter of the atlas canal reaches adult proportions at approximately age 5 or 6. Most of the congenital abnormalities encountered in the cervical spine involve the atlanto-axial segments. Vertebral synchondrosis are highly hazardous, because of their fragility, fractures and or dislocations can occur in connection with minute trauma [9,10,11].

Torticollis correlated with an underlying spine deformity has been described by Chiapparini et al. [12]. Conventional radiographs were of limited value, which were not attributable to the diagnosis of torticollis. Tomographic studies revealed the atlanto-axial rotatory fixation as the reason behind torticollis.

Ozer et al. [13] investigated the craniofacial and cervical spine anomalies in congenital muscular torticollis with three-dimensional computerised tomography. They measured the anatomical craniofacial dimensions via tomographic studies in six patients. Neither Chiapparini et al. or Ozer et al. had approached a definitive clinical diagnosis.

Spondylomegepiphyseal dysplasia has been described by Silverman and Reiley [2] as a condition affecting the spine, epiphyses and metaphyses of the long bones. The trunk and neck were short (the neck was also stiff) and there was defective ossification of the vertebral bodies with the anterior halves being separated from the well-developed posterior elements. Synchondrosis among the neural arches, body, and odontoid process fuse at 3–6 years of age. After the age of 6, odontoid process fuses with the body and the neural arches.

Appendicular and axial deformities in patients with pseudoachondroplasia are diverse Limbs are usually short, and the joints are lax, which predispose to the development of early arthritis. Al Kaissi et al. [14] described a child with unusual morbid development of the atlanto-axial vertebral segments initiating life-threatening dolicho-odontoid (sub-clinical basilar invagination). Os odontoideum (OO) is a congenital anomaly of the axis (C2), defined as a smooth, detached ossicle of different size and shape. Os odontoideum is separated from the base of a shortened odontoid process by an apparent gap, with no osseous connection to the body of the axis. The complications of atlanto-axial instability are vast and lead to serious myelopathic complications and may cause sudden death [15].

MURCS association is a combination of uterine aplasia/hypoplasia, renal agenesis/ectopy, abnormal cervical or upper thoracic vertebrae, abnormal ribs, Sprengel shoulder, upper limb abnormalities and deafness. Thus, the patients have features of Klippel-Feil anomaly with Rokitansky/urogenital adysplasia syndromes. It is postulated that the association stems from alteration of the blastemas of the upper somites, arm buds and pronephric ducts, which have an intimate relationship at the end of the fourth week of foetal life [16,17,18]. Sprengel’s deformity of the shoulder is a congenital anomaly of the shoulder girdle that is accompanied with tremendous limitations of the physiological and anatomical downward movement of the scapula [19].

The omovertebral bone (defined as a congenital defect of the cartilaginous or fibrous band connecting the vertebral border of the scapula with the cervical spine) gives rise to the development of Sprengel´s shoulder. In some, it may form a true joint with a small osseocartilaginous protuberance projecting from the mid-vertebral border to the scapula. The shoulder girdle musculature is usually defective with the trapezius being most often affected. The muscle may be absent or weak, especially in its lower portion. The rhomboids and elevator scapulae are usually hypoplastic and partially fibrosed. The serratus anterior may be weak, the pectoralis major, pectoralis minor, latissimus dorsi and sternocleidomastoid muscles may be affected [19]. Al Kaissi et al. [20] described vertebral segmentation defects in three family subjects. Sprengel´s anomaly, omovertebral bones, and segmentation defects of the vertebral column at the cervical, thoracic and sacral level were the obvious orthopaedic abnormalities. Al Kaissi et al. [21] described a girl and her mother with specific facial dysmorphic features associated with distinctive spine abnormalities in a girl and her mother.

## 10. Conclusions

In clinical practice, children/adults manifesting backbone maldevelopment represent a rich area for unlimited research and knowledge. Unfortunately, it has been customary for the majority of paediatric and orthopaedic practices to simply use the term ‘idiopathic’ to designate a long list of undiagnosed syndromic entities. In the vast majority of medical disciplines, only in the presence of craniofacial dysmorphic features can a syndrome be implied. Frankly speaking, true syndromic entities are rare and appear only in severely affected patients. Nevertheless, the overall skeletal pathologies encountered in orthopaedic departments are so common that more than 40% are genetically determined in patients with normal facial and intellectual features.

Torticollis is a major malformation complex, requiring a comprehensive clinical documentation. Physical therapy and stretching exercises organised for children with congenital torticollis in connection with syndromic association could lead to negative and possibly grave outcomes. Every patient should be documented on the bases of a thorough clinical phenotypic characterisation. In our group of patients, torticollis has been presented as a symptom complex rather than a sole diagnostic entity. The interpretation of cranio-cervical radiographs should be assessed in conjunction with a proper understanding of the craniofacial/spine anatomy. Upper cervical spine abnormalities are considered the most important and require prompt and distinctive measures through immobilisation and neurosurgical management.

The aetiological diagnosis and presentations of skeletal disorders highly overlap and are genetically very heterogeneous, which can be challenging to diagnose.

## Figures and Tables

**Figure 1 diagnostics-12-01672-f001:**
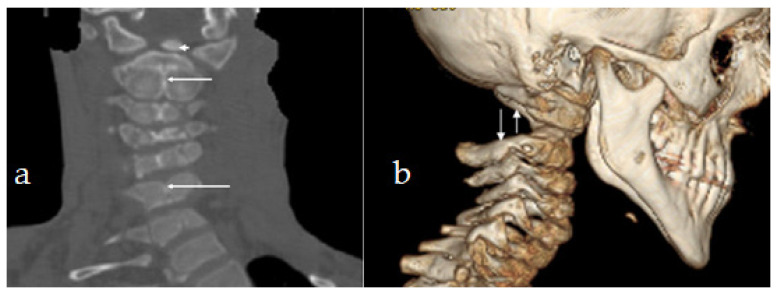
(**a**). Reformatted Coronal CT scan of the cervical spine showed butterfly vertebrae (defective formation), note the detached cephalic part of the odontoid process (arrowhead) in connection with extensive cervical spine synchondrosis causing the mal-development of butterfly vertebrae. (**b**) Reconstruction CT scan of the cervical spine showed atlanto-axial dislocation (arrows).

**Figure 2 diagnostics-12-01672-f002:**
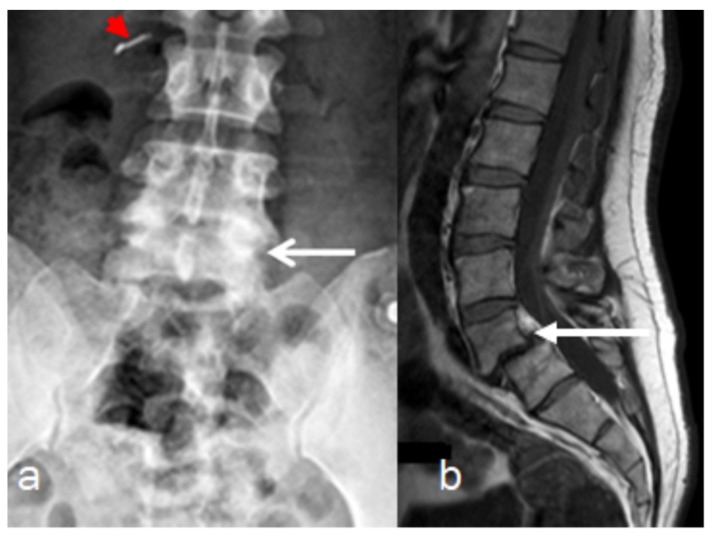
(**a**,**b**). Imaging of the parents, AP radiograph of the lumbar vertebrae of the 32-years-old-mother showed osteoarthritis of the facet joints of L5 (arrow) associated with diminution of the heights of L4/5. Note the increased level of calcification that signifies facet arthritis. Surprisingly, incidental diagnosis of early ossification of the abdominal aorta-red arrow head (**a**). Sagittal spine MRI of the 38-year-old father showed dysplastic spondylolisthesis of L4/5 (arrow) that might leads to spinal stenosis (**b**).

**Figure 3 diagnostics-12-01672-f003:**
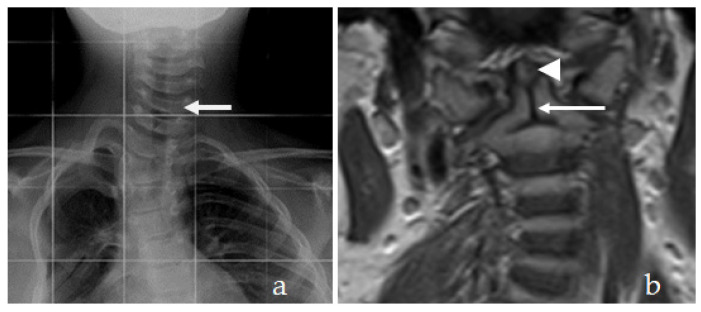
(**a**,**b**) AP cervical and upper thoracic spine of a 16-year-old girl with pseudoachondroplasia showed torticollis associated with unusual flattening of the cervical spine (arrow). The MRI has illustrated odontoid dysplasia (arrow head) the persistence of dentocentral synchondrosis (arrow). The localisation and level of the remnant of the dentocentral synchondrosis started from the detached odontoid process and extends downwards to involve the entire cervical spine.

**Figure 4 diagnostics-12-01672-f004:**
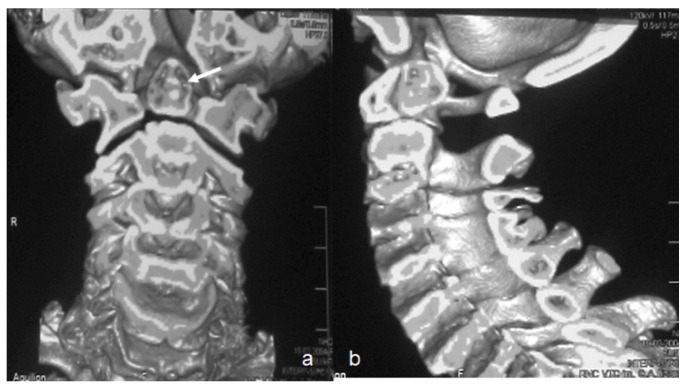
(**a**,**b**) Three-dimensional reconstruction coronal CT scan of a 14-year-old girl with pseudoachondroplasia showed rounded shape os odontoideum (arrow) and atlanto-axial instability (**a**). 3D sagittal reconstruction CT showed exaggerated cervical spine lordosis and atlanto-axial instability (**b**).

**Figure 5 diagnostics-12-01672-f005:**
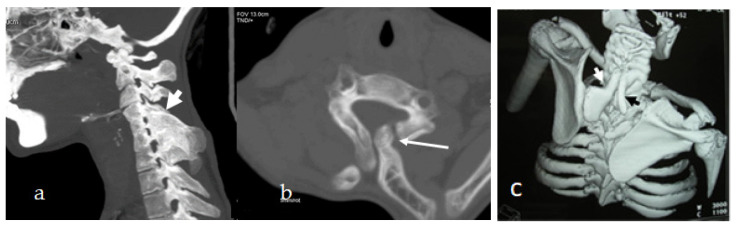
(**a**,**b**) Three-dimensional sagittal CT scan of an 18-year-old-girl with MURCS association showed cervical kyphosis with massive fusion of the cervical vertebrae (extensive fusion of C4-7 associated with fused spinous processes), a deformity that is extremely surprising (**a**). Axial 3D reformatted CT scan of the C7 showed true bony bridge between the omoplate and spine that resulted in the development of omovertebral bone (**b**). Three-dimensional reconstruction CT scan in a 16-year-old girl presented with torticollis, progressive cervico-thoracic kyphosis and bilateral omo-vertebral bones originated from the 7th cervical spine. Apparent omovertebral bone connecting the 7th cervical spine with the superior border of the left scapula (white arrow) gives rise to the development of Springle’s deformity (**c**).

**Table 1 diagnostics-12-01672-t001:** Phenotype/genotype of the family with spondylo-meg-epiphyseal-dysplasia.

Family Subject	Clinical Presentation	Imaging	Diagnosis	Genotype
Proband-16 years	Torticollis	Cartilaginous spine	Spondylo-meg-epiphyseal dysaplasia	Homozygous inactivating mutations in the NKX3-2 gene
12-years-sibling	---	No deformities	Genetic carrier	Heterozygous
Mother 32 years	Backpain	Osteoarthritis of the facet joints of L5 in connection with spondylolisthesis of L4-5. Incidental diagnosis of ossified abdominal aorta.	Genetic carrier	Heterozygous
Father 38 years	Backpain	Degeneration of L4/5 (Hernial disc prolapse in correlation with spondylolisthesis of L4-5	Genetic carrier	Heterozygous

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
