# Peer review of "Torticollis in Connection with Spine Phenotype"

_diagnostics, 2022, doi:10.3390/diagnostics12071672_

Round 1

Reviewer 1 Report

I think the manuscript is well written and can be accepted after some major language revision.

Author Response

Dear Reviewer,

We edited our MS accordingly.

Best regards

Reviewer 2 Report

A very confusing manuscript, without a clearly defined methodology and results. Discussion without contemporary literary citations. The literature is not written correctly, not a single quote from the paper that has been published in the last 5 years.

Author Response

Dear Reviewer,

We explained the nature of our work.

Best regards

Reviewer 3 Report

Type of the article is case series

line 92 - please add the name of the author you cited (3)

Lines 45, 50, 57, 67, 83, 87, 94, 97, 104, 126, 139, 155, 208, 213, 230, 249, and 262 should start with a tabulation

there is a lack of transition between firt part of intruductona and the part with descriptoion of potential causes of torticollis. Why did you choose these syndromes?

Material and method

how many patient were included to this paper?

what were the inclusion criteria for these patiets? how many patient with torticolis you treaoted in your Institute?

what was the protocol of clinical and radiological assesment for the patients? DNA analysis should be shortly described.

line 117 - please erase the bracket before Ilizarov Scientific Research Institute

Table 1 - age of parents does not correspond to data included in the patient description

The second described patient with MURCS reject doing genetic testing, so how do you know it's really this syndrome?

Discussion is a simple repeaing of introduction (description of syndromes). There is no discussion with literature data. What are the other possible rare causes of torticollis? 

references - all references are older than 10 years and also 5 out of 21 articles are self-citation -need to be refreshed

Author Response

Dear Reviewer,

Thank you so much for your points. We almost follow most of it.

Best regards

Round 2

Reviewer 2 Report

Torticollis in connection with Spine phenotype and Genotype

The value of this paper is to sensitize 72 physicians and orthopedic surgeons to the necessity of comprehensive clinical and radiological phenotypic 73 characterizations in patients with long term skeletal patholo…

If the title of the paper is “Torticollis in connection with Spine phenotype and Genotype”, then the significance is on this clinical entity.

Materials and Methods

This study was 121 conducted based on a clinical and radiographic evaluation of a group of children

…. in nine patients (3 boys and 4 girls with age range of 2-14 years 50 and two parents).

You wrote that the age of children is 2-14 years, and further in the text the children are aged 16 and 18 ???

3. Torticollis in connection with Pathologic aberration of the spine cartilaginous stage in a family with 134 Spondylo-meg-epiphyseal dysplasia

Torticollis was the prominent orthopaedic deformity in nine patients (3 boys and 4 girls with age range of 2-14 years 50 and two parents)…..

….A 16-year-old boy, with Marfanoid habitus, long and thin limbs and dysmorphic features. Torticollis associated with 136

5. Torticollis associated with odontoid abnormalities and atlanto-axial instability in patients 215 with pseudoachondroplasia

AP cervical and upper thoracic spine of a 16-year-old –girl with pseudoachondroplasia showed torticollis associated 222 with cervical spine platyspondyly.

A 14-year old girl manifested the phenotype and genotype of pseudoachondroplasia, presented with torticollis. 239 Radiological and tomographic studies confirmed the suspicion of os odontoidium. 240

7. Torticollis in patients with MURCS association 255

 An 18-year old girl with normal secondary sex characteristics, but primary amenorrhoea and cervical vertebral defects 256

A16-year old girl similarly presented with torticollis since her early life followed by the development of cervico-thoracic 269

Results

4. Sequence analysis of the NKX3-2 (BAPX1) gene revealed a homozygous 19 base pairs duplication (c.- 195 2_17dupAGATGGCTGTGCGCGGCGC) in the affected child. Parents were heterozygous carriers. The duplicated 196 region includes the initiation codon, which could lead to the production of a truncated protein. The identified variant 197 has not been previously reported in the databases nor in the literature.

(12-yrs-sibling have no torticollis)

6. Sequence trace analysis revealed that the patient was heterozygous for the nucleotide change c.950A>G in exon 9 of the 251 COMP gene. This mutation is predicted to result in the amino acid substitution p.Asp317Gly in the 1st type III repeat of 252 COMP (T31).

7.1. Results???? (It should be written 8)

8. The 18-year-old girl underwent chromosomal analysis showed a balanced translocation of 46 XX, t (14q;15q). The 288 parents of the 16-year-old girl rejected any additional laboratory or genetic testings.

In the methods you state that you have nine patients ( 3 boys and 4 girls, two parents) and in the results you have 6 patients ???

9. Discussion

Torticollis in connection with Spine phenotype and Genotype is the title of the paper, and the point of the discussion is the existence of torticollis, types, significance and prevalence in syndromic pathology. It was said about torticollis ..

Torticollis correlated with an underlying spine malformation complex has been described by Chiapparini et al (12). 313 Conventional radiographs were of limited value and were cervical radiographs examinations, which were not attributa- 314 ble to the diagnosis. Tomographic studies revealed the reason. Ozer et al (13) investigated the craniofacial and cervical 315 spine anomalies in congenital muscular torticollis with three-dimensional computerized tomography. 316

Conclusion

 If the title of the paper is Torticollis in connection with Spine phenotype and Genotype, then the significance is on this clinical entity.

Torticollis has been presented as a symptom complex rather than 364 a sole diagnostic entity…

Can you explain?

References

Briggs MD, Mortier GR, Cole WG, et al. Diverse mutations in the gene for cartilage oligomeric matrix pro-  tein in the Pseudoachondroplastic-multiple epiphyseal dysplasia disease spectrum. Am J Hum Genet  1998;62:311-319. Hum Genet 1998;103:633-638.

Ikegawa S, Ohashi H, Nishimura G, Kim KC, et al. Novel and recurrent COMP (cartilage oligomeric matrix 418 protein) mutations in pseudoachondroplasia and multiple epiphyseal dysplasia?????

Mahajan P, Kher A, Khungar A, et al. MURCS association - a review of 7 cases. J Postgrad Med 423 1992;38:109-111  (109-11).

Biason-Lauber, A., Konrad, D., Navratil, F., et al. (2004). A WNT4 mutation associated with Müllerian-Duct 425 regression and virilization in a 46,XX woman. N. Engl. J. Med.????

Biason-Lauber, A., De Filippo, G., Konrad, D., et al. (2007). WNT4 deficiency-a clinical phenotype distinct 427 from the classic Mayer-Rokitansky-Kuster-Hauser syndrome: A case report. Hum. Reprod.????

……………

The literature is not written (corrected) correctly and there is not a single reference that has been published in the last 5  (10) years.

Author Response

Dear Reviewer,

Thank you for your comments. We followed your comments point by point

Best regards

Al Kaissi

The value of this paper is to sensitize 72 physicians and orthopedic surgeons to the necessity of comprehensive 
clinical and radiological phenotypic 73 characterizations in patients with long term skeletal patholo… 
 If the title of the paper is “Torticollis in connection with Spine phenotype and Genotype”, then the significance is on 
this clinical entity. 
The title modified: Torticollis in connection with Spine phenotype 
 Materials and Methods
This study was 121 conducted based on a clinical and radiographic evaluation of a group of children 
…. in nine patients (3 boys and 4 girls with age range of 2-14 years 50 and two parents). Corrected 
You wrote that the age of children is 2-14 years, and further in the text the children are aged 16 and 18 ??? Sorry, 
The youngest patient is 14 yrs- corrected 3. Torticollis in connection with Pathologic aberration of the spine cartilaginous stage in a family with 134 
Spondylo-meg-epiphyseal dysplasia
Torticollis was the prominent orthopaedic deformity in seven patients (3 boys and 4 girls with age range of 16-18 
years 50 and two parents)…..Corrected 
….A 16-year-old boy, with Marfanoid habitus, long and thin limbs and dysmorphic features. Torticollis associated 
with 136: Corrected 
5. Torticollis associated with odontoid abnormalities and atlanto-axial instability in patients 215 with 
pseudoachondroplasia
AP cervical and upper thoracic spine of a 16-year-old –girl with pseudoachondroplasia showed torticollis associated 
222 with cervical spine platyspondyly. Yes, unusual flattening of the cervical spine (CORRECTED) 
A 14-year old girl manifested the phenotype and genotype of pseudoachondroplasia, presented with torticollis. 239 
Radiological and tomographic studies confirmed the diagnosis of os odontoidium. 240- corrected 
7. Torticollis in patients with MURCS association 255 
 An 18-year old girl with normal secondary sex characteristics, but primary amenorrhoea and cervical vertebral 
defects 256- CORRECTED 
A16-year old girl similarly presented with torticollis since her early life followed by the development of cervicothoracic 269- Progressive cervico-thoracic kyphosis (related to growth spurt) 

Results
4. Sequence analysis of the NKX3-2 (BAPX1) gene revealed a homozygous 19 base pairs duplication (c.- 195 
2_17dupAGATGGCTGTGCGCGGCGC) in the affected child. Parents were heterozygous carriers. The duplicated 
196 region includes the initiation codon, which could lead to the production of a truncated protein. The identified 
variant 197 has not been previously reported in the databases nor in the literature. The proband –corrected 
(12-yrs-sibling have no torticollis)- He showed no skeletal deformities. 

6. Sequence trace analysis revealed that the proband was heterozygous for the nucleotide change c.950A>G in exon 
9 of the 251 COMP gene. This mutation is predicted to result in the amino acid substitution p.Asp317Gly in the 1st 
type III repeat of 252 COMP (T31). 

7.1. Results???? (It should be written 8)-CORRECTED 
8. The 18-year-old girl underwent chromosomal analysis showed a balanced translocation of 46 XX, t (14q;15q). The 
288 parents of the 16-year-old girl rejected any additional laboratory or genetic testings. The rejection comes from 
her parents. 

In the methods you state that you have nine patients ( 3 boys and 4 girls, two parents) and in the results you have 6 
patients ??? 
1. Spondylo-meg-epiphyseal dysplasia: Four family subjects 
2. Pseudoachondroplasia : Two unrelated patients 
3. MURCS : Two unrelated patients 
The total is 6 in addition to a couple of parents, the total is 8 subjects -sorry 
9. Discussion
Torticollis in connection with Spine phenotype and Genotype is the title of the paper, and the point of the 
discussion is the existence of torticollis, types, significance and prevalence in syndromic pathology. It was said about 
torticollis .. Changed 
Torticollis correlated with an underlying spine malformation complex has been described by Chiapparini et al (12). 
313 Conventional radiographs were of limited value and were cervical radiographs examinations, which were not 
attributa- 314 ble to the diagnosis. Tomographic studies revealed the reason. Ozer et al (13) investigated the 
craniofacial and cervical 315 spine anomalies in congenital muscular torticollis with three-dimensional computerized 
tomography. 316: CORRECTED 

Conclusion
 If the title of the paper is Torticollis in connection with Spine phenotype and Genotype, then the significance is on 
this clinical entity. The title has been restricted to the clinical aspects of our patients 
Can you explain?
Torticollis in our patients has been presented as a symptom complex rather than a sole diagnostic entity…The 
traditional diagnosis for torticollis in the vast majority of orthopedic practices is strongly correlated to muscular 
contractions. Rarely, a comprehensive clinical and radiological phenotypic characterization are performed. Most of 
our colleagues realized the magnitude of false diagnosis only when the deformity become progressive. 
References
Briggs MD, Mortier GR, Cole WG, et al. Diverse mutations in the gene for cartilage oligomeric matrix pro- tein in 
the Pseudoachondroplastic-multiple epiphyseal dysplasia disease spectrum. Am J Hum Genet 1998;62:311-319. 
Hum Genet 1998;103:633-638. 
Ikegawa S, Ohashi H, Nishimura G, Kim KC, et al. Novel and recurrent COMP (cartilage oligomeric matrix 418 
protein) mutations in pseudoachondroplasia and multiple epiphyseal dysplasia?????
Mahajan P, Kher A, Khungar A, et al. MURCS association - a review of 7 cases. J Postgrad Med 423 1992;38:109-
111 (109-11). 
All the above mentioned references have been replaced 

Biason-Lauber, A., Konrad, D., Navratil, F., et al. (2004). A WNT4 mutation associated with MüllerianDuct 425 regression and virilization in a 46,XX woman. N. Engl. J. Med. 19;351(8):792-8. doi: 
10.1056/NEJMoa040533 
Biason-Lauber, A., De Filippo, G., Konrad, D., et al. (2007). WNT4 deficiency-a clinical phenotype distinct 427 
from the classic Mayer-Rokitansky-Kuster-Hauser syndrome: A case report. Hum. Reprod;22(1):224-9. 
 .???? CORRECTED 
……………
The literature is not written (corrected) correctly and there is not a single reference that has been published in the last 
5 (10) years. 
We selected the most compatible

Round 3

Reviewer 2 Report

I asked you to correct the entire literature, not just the first few references

. Chiapparini, L., Zorzi, G., De Simone, T., et al. (2005). Persistent fixed torticollis due to atlanto-axial rotatory 442 fixation: Report of 4 pediatric cases. Neuropediatrics. doi:10.1055/s-2004-830533. doi:10.1002/(SICI)1096- 443 8628(19960301)62:1<_x0031_:_x003a_AID-AJMG1>3.0.CO;2-1.

Al Kaissi, A., Ganger, R., Hofstaetter, J. G., et al. (2011). The aetiology behind torticollis and variable spine 457 defects in patients with Müllerian duct/renal aplasia-cervicothoracic somite dysplasia syndrome: 3D CT scan 458 analysis. Eur. Spine J. doi:10.1007/s00586-011-1835-1.

Moradkhani, K., Puechberty, J., Bhatt, S., et al. (2006). Rare Robertsonian translocations and meiotic 460 behaviour: Sperm FISH analysis of t(13;15) and t(14;15) translocations: A case report. Hum. Reprod. 461 doi:10.1093/humrep/del314.

Al Kaissi, A., Chehida, F. Ben, Gharbi, H., et al. (2005). Familial vertebral segmentation defects, Sprengel 465 anomaly, and omovertebral bone with variable expressivity. Am. J. Med. Genet. doi:10.1002/ajmg.a.30968.

Author Response

Dear Reviewer,

Thank you so much for your observations. The literature has been corrected accordingly. 

Best regards

Al Kaissi
